# Molecular Characteristics, Receptor Specificity, and Pathogenicity of Avian Influenza Viruses Isolated from Wild Ducks in Russia

**DOI:** 10.3390/ijms231810829

**Published:** 2022-09-16

**Authors:** Elizaveta Boravleva, Anastasia Treshchalina, Yulia Postnikova, Alexandra Gambaryan, Alla Belyakova, Galina Sadykova, Alexey Prilipov, Natalia Lomakina, Aydar Ishmukhametov

**Affiliations:** 1Chumakov Federal Scientific Center for the Research and Development of Immune-and-Biological Products, Village of Institute of Poliomyelitis, Settlement “Moskovskiy”, 108819 Moscow, Russia; 2Department of Virology, Faculty of Biology, Lomonosov Moscow State University, 119991 Moscow, Russia; 3The Gamaleya National Center of Epidemiology and Microbiology of the Russian Ministry of Health, 123098 Moscow, Russia

**Keywords:** avian influenza, pathogenicity

## Abstract

Avian influenza viruses (AIV) of wild ducks are known to be able to sporadically infect domestic birds and spread along poultry. Regular surveillance of AIV in the wild is needed to prepare for potential outbreaks. During long-year monitoring, 46 strains of AIV were isolated from gulls and mallards in Moscow ponds and completely sequenced. Amino acid positions that affect the pathogenicity of influenza viruses in different hosts were tested. The binding affinity of the virus for receptors analogs typical for different hosts and the pathogenicity of viruses for mice and chickens were investigated. Moscow isolates did not contain well-known markers of pathogenicity and/or adaptation to mammals, so as a polybasic cleavage site in HA, substitutions of 226Q and 228G amino acids in the receptor-binding region of HA, and substitutions of 627E and 701D amino acids in the PB2. The PDZ-domain ligand in the NS protein of all studied viruses contains the ESEV or ESEI sequence. Although several viruses had the N66S substitution in the PB1-F2 protein, all Moscow isolates were apathogenic for both mice and chickens. This demonstrates that the phenotypic manifestation of pathogenicity factors is not absolute but depends on the genome context.

## 1. Introduction

Natural hosts of avian influenza viruses (AIVs) are the aquatic birds of the orders *Anseriformes* and *Charadriiformes*. AIVs with the 16 subtypes of hemagglutinin (HA) and with the nine subtypes of neuraminidase (NA) circulate in these hosts. The H5, H7, H6, and H9 subtypes of HA are found both in aquatic birds and in poultry [1]. Pigs, horses, and humans are other potential hosts for influenza viruses. In these hosts, the H1, H2, H3, and H7 subtypes of HA are usual [2,3].

In the primary hosts (waterfowl), the viruses replicate in the epithelial cells of the lower intestine, cause asymptomatic infection, and are transmitted by the fecal–oral route [4]. In new hosts, the way of transmission, cellular tropism, and pathogenicity may change. In chickens, the virus acquires the ability to spread throughout the body and evolves towards increased pathogenicity. This may be due to cannibalism being an effective way of transmitting the virus among chickens [5]. The AIVs of the H5 and H7 subtypes became fatal not only for birds but also for mammals, including humans.

As a rule, virus introduction into a new host is accompanied by a change of receptor-binding phenotype. Studies on viral receptor-binding specificity revealed that preferential binding to terminal Neu5Ac2-3Gal disaccharide is shared by the majority of avian viruses; however, viruses adapted to humans, and sometimes, pigs recognize the Neu5Ac2-6Gal terminated receptors [6,7]. Recognition of Neu5Ac2-6Gal receptors is a necessary prerogative of a pandemic virus since such determinants are widely present in the human nonciliated cells of the upper respiratory tract and the corresponding tropism of the virus determines the ability to airborne transmission [8,9]. However, the human infections caused by H5N1 chicken viruses demonstrated that avian viruses with Neu5Ac2-3Gal receptor-binding specificity could infect humans and even cause fatal diseases [10]. This is because terminal Neu5Ac2-3Gal disaccharide is widely present in the pulmonary alveoli and in the tracheal ciliated epithelium [8].

All human cases of infections with AIV were caused by viruses of terrestrial poultry rather than by feral bird viruses. A comparison of influenza viruses from ducks, chickens, and gulls revealed differences in their receptor specificity. Duck viruses of various HA subtypes are preferentially bound to receptors having the α1-3 linkage between the terminal Neu5Ac2-3Gal moiety and the penultimate sugar residue. Sulfation at the 6-OH group of the sub-terminal GlcNAc had little effect on the binding of duck viruses, whereas fucosylation of this residue reduced the binding significantly [11]. H5N1 chicken viruses differed from H5 duck viruses by extraordinarily high affinity to sulfated trisaccharide Neu5Acα2-3Galβ1-4(6-HSO3)GlcNAcβ [12]. Some of the poultry H9N2 viruses acquired receptor specificity, which is similar to the receptor specificity of human influenza viruses [13]. Viruses isolated from gulls showed high-avidity binding to fucosylated sialyloligosaccharides [14].

Change in the receptor specificity of viruses is determined by mutations in the region of the receptor-binding site of hemagglutinin. The six amino acids in the HA receptor-binding site are highly conserved among avian viruses but bear substitutions in human viruses (A138S, E190D, L194I, and G225D for H1 HA or Q226L and G228S for H2 and H3 HA) [15]. Substitutions X222Q and S227R enhanced binding to fucosylated sialylologosaccharides during the adaptation of the virus to gulls [14,16]. 193K is typical for H5 viruses. It can be speculated that the interaction of positively charged 193K with the 6-O-Su-group of glucosamine enhances the binding with sulfated receptor [11] (See also Figure 1).

Receptor specificity is not the only factor associated with the host range of the virus. The PB2 subunit of the viral polymerase is a host range determinant, and the PB2 mutations E627K and D701N play an important role in virus adaptation to mammalian hosts [17,18,19]. RNA polymerases of AIVs replicate viral RNA inefficiently in human cells because of species-specific differences in acidic nuclear phosphoprotein 32 (ANP32). Host-adaptive mutation E627K of the PB2 subunit enables polymerases to overcome this restriction and efficiently replicate viral RNA in the presence of human ANP32 [20]. Mutation D701N combined with mutation S714R is involved in adaptation to mammals through exposing a nuclear localization signal that mediates importin-α binding and entry of PB2 into the nucleus, where the virus replicates [21].

As we mentioned above, a change in the host may be accompanied by an increase in the pathogenicity of the virus. The pathogenicity of IVs has a multigenic character [22]. The key pathogenicity factor for H5 and H7 AIVs is the polybasic cleavage site of HA, which is cleaved by furins (intracellular serine proteases of animal cells). This type of cleavage allows the virus to multiply in the internal organs and leads to fatal systemic infection, in contrast to local infection caused by low pathogenic influenza viruses due to the limited cleavage of HA at monobasic cleavage sites.

The PB1 gene may encode an additional PB1-F2 protein that promotes virulence by inducing apoptosis of infected cells. The PB1-F2 66S variant reduces the production of interferon (INF) [23]. N66S substitution of PB1-F2 is partly responsible for the high virulence of highly pathogenic influenza viruses (HPAI) H5N1 [24].

The NS1 protein functions as an antagonist to block the type 1 INF-mediated host antiviral response [25]. D92E in NS1 provided resistance to the antiviral effects of INFs and increased the virulence of HPAI H5N1 in mice and in pigs [26,27].

Four terminal amino acids of the NS1 protein form a PDZ ligand domain of the X-S/T-X-V type. This PDZ ligand domain of NS1 has also been shown to influence influenza virus virulence. Pandemic 1918 H1N1 and H5N1 HPAI viruses contain a PDZ ligand domain motif, which increases virulence when introduced into a mouse-adapted influenza strain [28].

It is generally accepted that wild duck IVs are safe for humans and other mammals. At the same time, cases of infection of mammals with viruses of poultry were noted many times. Outbreaks of highly pathogenic HPAI (H5) viruses in 2022 have been repeatedly observed in Asia, Europe, and Africa. Many cases of human infection with H5N1/N6 viruses as well as H9N2 viruses have been noted. Cases of infection of a wild red fox with HPAI H5N1 have been noted in the USA and Canada [29].

Epidemiological surveillance of influenza viruses in the wild and in communities is essential to maintain preparedness for new pandemics.

Since 2006, we have been studying the diversity of AIVs carried by gulls and mallards in Moscow ponds. The purpose of this study was to an assessment of the potential danger of these viruses in the possibility of transmission to humans or pets. Based on complete genome sequencing, the key factors of host range and pathogenicity were analyzed. Receptor binding specificity was determined, and the pathogenicity of viruses for mice and chickens was investigated.

## 2. Results

During the autumn periods of 2006–2021, nearly 3000 samples of feces from gull (*Larus ridibundus*) and mallard (*Anas platyrhynchos*) were collected on the banks of the Moscow ponds, and AIVs were isolated. The isolates were subtyped by PCR with specific primers and sequenced. All viral genes of 46 strains were fully sequenced (GenBank accession numbers are given in Appendix A), and four strains were partly sequenced (Appendix A).

### 2.1. Molecular Characteristics

All sequenced strains were analyzed for factors of host range and virulence. Basic amino acids crucial for IV pathogenicity are shown in Table 1. The key pathogenicity factor for influenza viruses is the primary structure of the HA proteolysis site. In all studied sequences, the site of proteolysis has one arginine and corresponds to the consensus sequences of apathogenic viruses for each HA serotype. The receptor-binding site contains 226Q and 228G in HA of all studied viruses, which is typical for avian influenza viruses and is associated with the recognition of receptors terminated by sialyl(2-3)galactose groups [6,7]. In position 222, which is critical for the recognition of fucosylated sialosugars [11], there are bulky amino acids (K or W) in all isolates but H6 ones that contain alanine in this position (Figure 1).

The model represents the crystal structure of the H5 HA complex with LSTa [30], in which sialyloligosaccharide was modified to generate Su-SLe^x^ using Discovery Studio ViewerPro5.0 (Accelrys Inc., San Diego, CA, USA) software. The HA is shown as a grey molecular surface. 193K and 222K are highlighted and labeled. The fucose residue of Su-SLe^x^ colored brawn, the sulfo group, is shown in orange-red. The image illustrates the interposition of sulfo group and 193K as well as fucose and 222K.

Host-specific amino acids 627 and 701 in the PB2 were also analyzed. All Moscow isolates have avian host-specific 627E. The D701N substitution determining the preferred binding with human importin-α was not noted.

The ligand of the PDZ domain has the ESEV sequence in all NSs, except for the d/4031/2010 virus, with the ESEI sequence. The first variant is the most common among avian influenza viruses, and the second is less common but also characteristic of avian influenza [31].

In position 66 of the additional viral protein PB1-F2, most of the isolates have N, although ten cases of N66S and one case of N66T substitutions were noted.

### 2.2. The Receptor Specificity of Moscow AIVs in Comparison with Duck, Gull, Poultry, Swine, and Human Influenza Viruses

Receptor specificity of the viruses was characterized by determining their affinity to soluble sialylglycopolymers in a competitive assay based on the inhibition of the virus binding with labeled fetuin [32]. The data were expressed in terms of concentration of 50% inhibition (Ing_50_). For the calculation of Ing_50_, the concentration of the sialic acid residues in the solution was used. The structures and designations of the oligosaccharide moieties are shown in Table 2.

To compare the receptor-binding specificity of Moscow isolates with influenza viruses of different hosts, we provided parallel testing of 11 of our strains with reference duck, gull, chicken, swine, and human influenza viruses.

The binding of viruses to six distinct polymeric glycoconjugates was determined in a competitive binding assay (see Table 2 for structural formulas). One of the glycoconjugates (6′SLN) harbored 6-linked sialyloligosaccharide. The oligosaccharide parts of the glycopolymers with terminal Neu5Acα2-3Gal moiety differed by the type of the bond between galactose and the next sugar residue (β1-3 or β1-4) and by substituents at different positions on the GlcNAc ring (fucose or/and sulfo group). The values of Ing_50_, which reflect the affinity of the virus with the sialylglycopolymer (SGP), are given in Table 3.

The patterns of receptor-binding specificity varied significantly among reference viruses of different hosts. Only human virus has a high affinity with 6-linked sialyloligosaccharide. Duck virus has a high affinity with SLe^c^ and a low affinity with fucosylated SLe^x^ and Su-SLe^x^. Gull virus, contrary, has a maximal affinity with fucosylated moieties. Chicken viruses have a maximal affinity with sulfated sialosaccaride, while swine viruses—with sulfated and fucosylated Su-SLe^x^.

Moscow isolates demonstrate two patterns of binding. The binding with 6-linked sialyloligosaccharide was negligible for all strains. All tested strains of H1, H3, H4, H5, and H11 subtypes, similar to reference duck virus, have a maximal affinity with SLe^c^ and low affinity with fucosylated sialosaccarides.

At the same time, H6N2 viruses have an equal affinity with SLe^c^ and fucosylated SLe^x^. We detected such affinity patterns earlier for American and Asian H6 viruses. It was speculated that amino acids in position 222 (bulky or small) influence the sterical accommodation of the fucose into the receptor binding site [11]. The new data are consistent with this hypothesis, as all H6 viruses have 222A, while all other viruses have the bulky amino acid at this position (See Figure 1 and Table 1).

### 2.3. Pathogenicity of Viruses for Mice and Chickens

Moscow isolates did not contain markers of pathogenicity and/or adaptation to mammals in HA and PB2 gene segments. However, all studied viruses contain the terminal ESEV or ESEI sequence in the NS protein, and several strains had the N66S substitution in the PB1-F2 protein (Table 1). To explore the possible influence of these factors on the virulence of viruses, we tested 25 isolates for pathogenicity in chickens and 36 viruses for pathogenicity in mice (Table 4). The 30-day-old chickens infected orally with the tested viruses showed no signs of disease despite a pronounced immune response.

Balb/c mice weighing 12–14 g did not die after intranasal administration of high doses of virus (10^5^ EID_50_ per mouse), although weight loss was observed in some cases. To compare the pathogenicity of different strains, we chose such a characteristic as the ratio of weight on the fourth day after infection with a dose of 10^5^ EID_50_ per mouse (weighing 12–14 g) to weight on the day of infection. Mice infected with the H6N2, H11N9, and H11N6 subtypes and some of the H4N6 viruses continued to gain weight after infection and showed no signs of disease.

At the same time, H1N1/N2, H3N1/N2/N6/N8, H5N3, and some of the H4N6 viruses under such conditions of infection caused weight loss by 5–15% on the fourth day after infection (Table 4). We did not find significant differences in the pathogenicity of viruses with N versus S in position 66 PB1-F2.

All tested viruses, even in the absence of symptoms of the disease, stimulated the production of specific antibodies.

## 3. Discussion

In 2016, the highly pathogenic Asian viruses of the H5 subtype appeared in Europe [33]. Since then, outbreaks of H5 in Europe have repeated regularly. This highlights the need to monitor AIV in nature.

Since 2006, we have been studying the diversity of AIVs carried by wild waterfowl on five ponds in Moscow. Until 2013, we isolated up to nine strains of different subtypes with different genomic compositions annually. In subsequent years, the number and variety of isolated viruses decreased for reasons unknown to us.

Phylogenetic analysis showed that almost all the genes of the studied viruses belong to the pool of apathogenic duck viruses, although some of them are located on the branches of evolutionary trees that gave rise to chicken viruses in the past.

All viruses isolated from wild mallards and gulls were safe for birds and mammals. The genomes of these viruses did not contain such markers of pathogenicity and adaptation to mammals as a polybasic cleavage site in HA and substitutions of 226 and 228 amino acids in the HA receptor-binding site. Receptor-binding specificity of viruses was typical for AIV; only “avian” type receptors with 2–3 links between sialic acid and galactose ring were recognized.

Viruses of subtype H6, unlike all over viruses, have a high affinity with fucosylated receptors. We assume that this is due to the small amino acid 222A in hemagglutinin.

We did not detect substitutions of 627 and 701 amino acids in the PB2. The PDZ-domain ligand in the NS protein in all studied viruses has the sequence ESEV or ESEI, and several viruses have the N66S or N66T substitution in the additional viral protein PB1-F2. There have been reports that these factors increase the pathogenicity of viruses in mammals [28,34]. However, a comparison of the pathogenicity of closely related viruses with N66 and S66 in mice and chickens did not reveal a difference between them. In the context of the genomes of wild duck viruses, N66S substitution in PB1-F2 and terminal ESEV in NS protein were not associated with increased virulence—all Moscow isolates were apathogenic for both mice and chickens. The obtained results demonstrated that wild mallards on their autumn migration through Moscow are infected exclusively with non-pathogenic AIV. However, the situation can change at any time, and continued control of AIV in nature is essential.

## 4. Materials and Methods

### 4.1. Reagents

Fetuin and horseradish peroxidase were from Serva, Switzerland. Antibodies against mouse and chicken immunoglobulins conjugated with horseradish peroxidase were from Sigma-Aldrich, Inc., St. Louis, MO, USA. MycoKill AB solutions were from PAA Laboratories GmbH, Pasching, Austria. Viral RNA Mini Kit was from QIAGEN, Hilden, Germany. MMLV Reverse Transcription kit, random primers, nuclease-free water, DNA Ladder, and TAE buffer were from Evrogen, Moscow, Russia. Ribonuclease inhibitors were from Syntol, Moscow, Russia. Soluble synthetic polyN-(2-hydroxyethyl)acrylamide-based sialylglycopolymers (SGP) contained 20 mol% of specific sialyloligosaccharide attached to the 30-kDa polymer were from GlycoNZ, Auckland, New Zealand.

### 4.2. Viruses

Fresh feces of gulls and ducks were collected from 2006 to 2021 on the shore of the ponds in Moscow city. Feces were suspended in a double volume of phosphate-buffered saline (PBS) supplemented with 0.4 mg/mL gentamicin, 0.1 mg/mL kanamycin, 0.01 mg/mL nystatin, and 2% MycoKill AB solution. The suspension was centrifuged for 10 min at 4000 rpm, and 0.2 mL of the supernatant was inoculated into 10-day-old chicken embryos (CE). Infected allantoic fluid (IAF) was collected after 48 h and tested by hemagglutination assay with chicken erythrocytes. The virus amount was expressed in hemagglutinating units, and the 50% infective dose (EID_50_) for each virus stock was determined by titration in CE. All strains are stored in the virus repository of the Chumakov scientific center (Moscow, Russia). Full names, designations of the viruses, and GenBank accession numbers are given in Appendix A.

### 4.3. Competitive Binding Assay

Receptor specificity of influenza viruses was evaluated in a competitive assay based on the inhibition of binding to a solid-phase immobilized virus with bovine fetuin labeled with horseradish peroxidase (HRP) [32]. Virus-containing fluids (0.1 mL) were added to wells of the plate precoated with fetuin. After overnight incubation at 4 °C, the plate was washed with PBS containing 0.05% Tween-20. Solutions of HRP-labeled fetuin with sialylglycopolymers (SGPs) were added (50 µL/well) and incubated for 1 h at 4 °C. Plate was washed fivefold, the peroxidase reaction was performed, and optical density was determined at 492 nm on Multiscan reader. Curves of binding were plotted, concentrations of 50% inhibition were calculated for every virus, and every sialylglycopolymer and patterns of binding of viruses for different sualosugars was compared.

### 4.4. Molecular Models

Atomic coordinates of the H5 HA complex with LSTa (1JSN) were obtained from Protein Data Bank [30]. The Su-SLe^x^ model was constructed on the basis of Sialyl Lewis^x^ (2KMB) [35] structure via replacing the hydrogen of the 6OH hydroxyl group of glucosamine with the HSO_3_ group. The fitting of Su-SLe^x^ into the receptor binding site of H5 HA was provided by superimposing the galactose residue of the sulfated analog over the galactose residue of LSTa. The molecular models were generated with the DS ViewerPro 5.0 software (Accelrys Inc., San Diego, CA, USA)

### 4.5. Animals

Chickens and embryonated chicken eggs were purchased from the State poultry farm “Ptichnoe” (Moscow, Russia). BALB/c mice were purchased from “Lesnoye” farm, Moscow, Russia. All studies with HPAIV viruses were conducted in a biosafety level 3 containment facility.

### 4.6. Ethics Statement

Studies involving animals were performed in accordance with the European Convention for the Protection of Vertebrate Animals used for Experimental and Other Scientific Purposes, Strasbourg, 18 March 1986. All appropriate measures were taken to ameliorate animal suffering. In total, 119 chickens were used in the study. Survived chickens were subsequently kept in the bird housing facility for antibody-level observation. The study design was approved by the Ethics Committee of the Chumakov Federal scientific center for the research and development of immune-and-biological products, Moscow, Russia (Approval #4 from 2 December 2014).

### 4.7. Infection of Mice

Six–ten-week-old BALB/c mice were used. Groups of mice were anesthetized and inoculated intranasally with IAF (10^5^ EID_50_ per mouse) or placebo (PBS). Survival and body weight following infection were monitored daily. On day 15 post-infection, serum samples were taken from survivor mice for antibody titration.

### 4.8. Oral Administration of the Viruses to Chickens

Thirty-day-old chickens were infected orally. After leaving the birds overnight without water, one drinking bowl per 3 birds was placed in each cage in the morning. Each bowl contained 30 mL of ten-time diluted IAF. The individual dose was about 10^6^ EID_50_.

### 4.9. Measurement of Antibodies against Influenza Viruses in Mouse and Chicken Sera

The levels of antibody were assessed by ELISA assay with anti-mouse or anti-chicken IgG, and 96-well plates (Nunc, MaxiSorp) were covered with fetuin. Then, 100 μL of IAFs was added to each well of the fetuin-coated microplates, kept overnight at 4 °C, then washed and blocked with 0.2% BSA solution in PBS, 1 h. The blocking solution was removed, 100 μL of buffer (0.1% Tween-20, 0.2% BSA on PBS) was added to the wells, on which the sera were titrated, starting from a dilution of 1:50. Wells without virus were used as blank. Sera from uninfected animals served as negative controls. Incubation was performed for 4 h at 4 °C. After washing, peroxidase-labeled antibodies against mouse or chicken immunoglobulins (Sigma-Aldrich, Inc., St. Louis, MO, USA) were added. It was incubated for 2 h; after washing, a color reaction with tetramethylbenzidine substrate solution was carried out.

### 4.10. Sequencing

Viral RNA was isolated from the allantoic fluid of infected chicken embryos with a commercial QIAamp Viral RNA mini kit (Qiagen, # 52904). Full-length viral genome segments were obtained by reverse transcription and PCR with specific terminal primers, MMLV, and Taq-polymerase (Alpha-Ferment Ltd., Moscow, Russia). The amplified fragments were separated by electrophoresis in 1–1.3% agarose gel and subsequently extracted from the gel with the Diatom DNA Elution kit (Isogene Laboratory Ltd., Moscow, Russia, # D1031). Sequencing reactions were performed with terminal or internal primers with the BrightDye™ Terminator Cycle Sequencing Kit v3.1 (Nijmegen, The Netherlands), followed by analysis on an ABI PRISM 3100-Avant automated DNA sequencer (Applied Biosystems 3100-Avant Genetic Analyzer, Foster City, CA, USA). The Lasergene software package (DNASTAR Inc., Madison, WI, USA) was used for the assembly and analysis of nucleotide sequences.

## 5. Conclusions

From 2006 to 2021, avian influenza viruses were monitored in wild gulls and ducks during autumn migration through Moscow. Forty-six AIV strains were isolated and fully sequenced. Influenza viruses did not contain markers of adaptation to mammals and were apathogenic for mice and chickens.

## Figures and Tables

**Figure 1 ijms-23-10829-f001:**
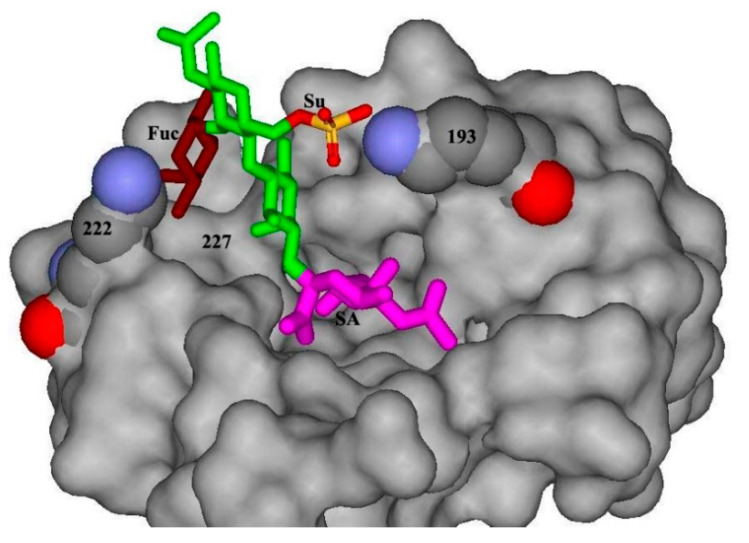
Molecular model of H5 HA complexed with sialyoligosaccharide Su-SLe^x^.

**Table 1 ijms-23-10829-t001:** Amino acids associated with increased pathogenicity of IAVs isolated in Moscow.

Virus Subtype and Strain Numbers *		Position in Proteins						
		PB2		PB1-F2	HA		NS	
		627	701	66	Cleavage Site	222–229 **	92	C-Term.
H1	4970, 5743, 5744	E	D	N	SIQSR-GLF	KVRGQAGR	D	ESEV
	5586, 5662	E	D	S	SIQSR-GLF	KVRGQAGR	D	ESEV
H3	3556, 3806, 4203, 4238, 4298, 4524, 4661, 4780, 4788, 5037, 5163, 5169, 5174, 5172, 5897, 5908	E	D	N	EKQTR-GLF	WVRGQSGR	D	ESEV
	4242, 4494, 4681	E	D	S	EKQTR-GLF	WVRGQSGR	D	ESEV
	5881	E	D	T	EKQTR-GLF	WVRGQSGR	D	ESEV
H4	3661, 3735, 3740, 3799, 4518, 4528, 4641, 4643, 4771, 4781, 4843	E	D	N	EKASR-GLF	WVRGQSGR	D	ESEV
	4652	E	D	S	EKASR-GLF	WVRGQSGR	D	ESEV
H5	4182, 4971	E	D	N	QRETR-GLF	KVNGQSGR	D	ESEV
	4952	E	D	N	QREAR-GLF	KVNGQSGR	D	ESEV
H6	3100	E	D	N	QIETR-GLF	AVSGQRGR	D	ESEV
	3720	E	D	N	QIETR-GLF	AVNGQRGR	D	ESEV
	4031	E	D	S	QIETR-GLF	AVNGQRGR	D	ESEI
H11	3641	E	D	N	AIASR-GLF	KVNGQAGR	D	ESEV
	5712	E	D	S	AIASR-GLF	KVNGQAGR	D	ESEV

* The viruses are designated by strain number; for example, 4970 stands for A/duck/Moscow/4970/2013. Full names of viruses are in Appendix A. ** HA amino acids are numbered according to H3. C-term.: C-terminus.

**Table 2 ijms-23-10829-t002:** Structures of oligosaccharides used in this study.

Saccharide	Abbreviation
Siaα (2-3) Galβ1-4GlcNAcβ	3′SLN
Siaα (2-3) Galβ1-4-(6-Su) GlcNAcβ	Su-3′SLN
(Siaα (2-3) Galβ1-4) (Fucα1-3) GlcNAcβ	SLe^x^
(Siaα (2-3) Galβ1-4) (Fucα1-3) (6-O-Su) GlcNAcβ	Su-SLe^x^
Siaα (2-3) Galβ1-3GlcNAcβ	SLe^c^
Siaα (2-6) Galβ1-4GlcNAcβ	6′SLN

**Table 3 ijms-23-10829-t003:** Concentration of sialoglycoconjugates provided 50% inhibition of binding viruses with HRP-labeled fetuin.

Viruses		Polymerized Receptor Analogs *					
		SLe^c^	3′SLN	Su-3′SLN	SLe^x^	Su-SLe^x^	6′SLN
Duck/Buryatiya/1905/2000	H4N6	10 **	20	20	100	100	>1000
Gull/Astrakhan/227/1982	H13N6	20	10	10	10	10	>1000
Chicken/NJ/294598-12/2004	H7N2	5	5	1	20	5	200
Chicken/HK/220/1997	H5N1	10	3	0.3	15	10	>1000
Swine/Kazakhstan/48/1982	H3N6	15	10	3	40	3	800
Swine/Hong Kong/9/1998	H9N2	>200	>200	2	>200	2	50
A/USSR/039/1968	H3N2	>200	>200	>200	>200	>200	4

Duck/4970/2013	H1N1	20	40	50	500	500	>1000
Duck/3556/2008	H3N1	4	10	10	200	200	>1000
Duck/3806/2009	H3N8	5	15	15	300	300	>1000
Duck/3661/2008	H4N6	5	10	8	700	500	>1000
Duck/3740/2009	H4N6	5	10	10	500	500	>1000
Duck/3799/2009	H4N6	5	10	10	500	500	>1000
Duck/4182/2009	H5N3	8	15	10	250	250	>1000
Gull//3100/2006	H6N2	10	30	30	10	20	>1000
Duck/3720/2009	H6N2	10	40	50	10	20	>1000
Duck/4031/2010	H6N2	10	40	50	10	20	>1000
Duck/3641/2008	H11N9	5	10	8	800	600	>1000

* The Ing50 (mkM Neu5Ac) of soluble receptor analogs were determined in a competitive binding assay. Lower values of Ing50 reflect stronger binding of virus to the receptor. ** Values are obtained by averaging the data of three experiments and are approximated to one or two significant digits in accordance with the confidence interval. Colors depict levels of binding for each individual virus: yellow—maximal binding; blue—weak binding.

**Table 4 ijms-23-10829-t004:** Pathogenicity and immunogenicity of isolates for mice and chickens.

Strain	Subtype	PB1-F2	In Mice		In Chicken	
		66 (N/S)	Weight *	AB **	Mort. ***	AB **
g/3100/06	H6N2	N	110 ± 5	312	0/5	957
d/3554/08	H3N1	N	90 ± 8	1953	0/10	2783
d/3556/08	H3N1	N	97 ± 4	1865	nd	nd
d/3661/08	H4N6	N	110 ± 4	1127	nd	nd
d/3641/08	H11N9	N	115 ± 4	735	nd	nd
d/3720/09	H6N2	N	108 ± 3	634	0/4	571
d/3735/09	H4N6	N	105 ± 6	1542	nd	nd
d/3740/09	H4N6	N	90 ± 10	2468	nd	nd
d/3799/09	H4N6	N	95 ± 8	1652	nd	nd
d/3806/09	H3N8	N	98 ± 6	946	nd	nd
d/4203/10	H3N8	N	95	1645	nd	nd
d/4238/10	H3N6	N	94	1852	nd	nd
d/4242/10	H3N8	S	92	2523	nd	nd
d/4298/10	H3N8	N	96	1193	0/5	1200
d/4182/10	H5N3	N	98	1748	0/10	1534
d/4206/10	H5N3	N	97	2213	0/10	928
d/4031/10	H6N2	S	102	406	0/5	432
d/4494/11	H3N8	S	89	3215	nd	nd
d/4681/11	H3N8	S	96	1656	0/3	2832
d/4521/11	H3N8	S	86	3594	nd	nd
d/4522/11	H3N8	S	93	2163	nd	nd
d/4518/11	H4N6	N	96	1352	nd	nd
d/4528/11	H4N6	N	100	962	0/3	1589
d/4661/11	H3N8	N	84	3193	0/3	4235
d/4641/11	H4N6	N	102	473	nd	nd
d/4643/11	H4N6	N	99	1042	nd	nd
d/4652/11	H4N6	S	103	520	nd	nd
d/4771/12	H4N6	N	nd	nd	0/2	2944
d/4772/12	H4N6	N	nd	nd	0/5	2154
d/4970/13	H1N1	N	93	1130	0/6	1714
d/5037/14	H3N8	N	91	824	0/5	1664
d/5163/15	H3N6	N	nd	nd	0/2	4390
d/5169/15	H3N6	N	nd	nd	0/2	3171
d/5172/15	H3N6	N	nd	nd	0/2	3044
d/5662/18	H1N2	S	83	1943	0/3	2546
d/5586/18	H1N2	S	81	1754	0/6	1724
d/5743/19	H1N1	N	79	963	0/3	2832
d/5744/19	H1N1	N	83	1145	0/6	3676
d/5712/19	H11N6	S	110	500	0/3	1260
d/5881/21	H3N2	T	105	1414	0/2	4234
d/5897/21	H3N8	N	102	2048	0/2	2054
d/5908/21	H3N8	N	nd	nd	0/2	3527
Control			114 ± 4	<50	0/10	<200

* Average weight of Balb/c mice on the 4th day after infection in % of the weight on the day of infection. There were 6–20 mice in every group. ** Geometric mean antibody titer in sera (ELISA). *** Number of dead/number of infected chickens by oral infection with 10^6^ of EID_50_. nd—no data.

## Data Availability

Not applicable.

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
