# Peer review of "Molecular Characteristics, Receptor Specificity, and Pathogenicity of Avian Influenza Viruses Isolated from Wild Ducks in Russia"

_ijms, 2022, doi:10.3390/ijms231810829_

Round 1

Reviewer 1 Report

The authors need additional analysis and discussion on the data generated to provide more interesting information to the readers 

Author Response

Dear reviewers! I am very grateful for your constructive comments on our work. Almost all of your suggestions have been accepted and changes included to the text of the manuscript. Specific answers to all your comments are in the attached file.

Reviewer 2 Report

 Boravleva et al described the characteristics of influenza virus strains in wild ducks isolated in Moscow from 2006 to 2021. The time span of more than ten years is of great significance to the epidemiological study of influenza virus. For the article, I have some questions.

In line 20, “for different hosts” should be revised to “in different hosts”.

In line 27, the significance of this study should be briefly described at the end of abstract.

In line 42, “chicken influenza viruses” should be revised to “avian influenza viruses” or “AIV”.

In line 90, “furins -” should be revised to “furins”.

In line 97, “HPAI”, the first occurrence should be the full name, and then the abbreviation

In line 123, are all viral genes sequenced? Or partial viral gene sequencing, such as PB2, HA, NS, etc? It shall be clearly indicated.

In 2.1 section of results, the evolutionary tree analysis of 46 isolates should be added to explain the virus variation.

In line 176, SGP, the first occurrence should be the full name, and then the abbreviation

In Table 3, is there any possible difference between polymerized receptor assays and natural receptors in blocking viral infection?

In Table 4, how many mice were challenged in each group?

Syntax modification: there are redundant spaces in some statements. These should be revised.

In discussion section, the significance of the research results and their possible reference value should be reflected at the end of the discussion.

The formats of some references are irregular. Modify the format of references according to the publication requirements.

Author Response

(The authors gave the same response as above.)
